# Geomorphological Approach to Cliff Instability in Volcanic Slopes: A Case Study from the Gulf of Naples (Southern Italy)

**Giuseppe Di Crescenzo [1], Nicoletta Santangelo [2] , Antonio Santo [1] and Ettore Valente [2,*]**

[1] Department of Civil, Constructional and Environmental Engineering, University of Naples Federico II, Piazzale Tecchio 80, 80125 Naples, Italy; giuseppe.dicrescenzo@unina.it (G.D.C.); antonio.santo@unina.it (A.S.)

[2] Department of Earth, Environmental and Resources Sciences, DISTAR, University of Naples Federico II, Via Cinthia 21, 80126 Naples, Italy; nicsanta@unina.it

\* Correspondence: ettore.valente@unina.it

**Abstract:** This paper deals with the problem of cliff stability and proposes a geomorphological zonation of a cliff using a sector of the Posillipo promontory (named the Coroglio-Trentaremi sea cliff, Italy), in the Campi Flegrei coastal area, as a case study. A detailed geological and geomorphological analysis was carried out, by combining field work with analysis of detailed scale topographic maps, orthophoto, and stratigraphical data from deep boreholes. Field and borehole data, together with structural data collected in seven different stations along the cliff, allowed us to derive six geological cross-sections and to reconstruct the complex stratigraphical and structural setting of the cliff. Geomorphological analysis focused on the detection of the main geomorphological factors predisposing to cliff instability. We selected the most significant factors and divided them into two groups: factors influencing landslide intensity and factors influencing cliff instability. Then, by means of a heuristic approach, we constructed a matrix that was used to derive a map showing the geomorphological zonation of the sea cliff. This map may enable to development of a reliable scenario of cliff instability and consequent retreat, which may be useful either to plan intervention works in the most critical areas or to organize prevention plans aimed at risk mitigation.

**Keywords:** volcanic coast; cliff instability; rock falls; Campi Flegrei; southern Italy

## 1. Introduction

Studying the evolution of coastal areas is a crucial issue because coastal areas have experienced human frequentation since historical and even pre-historical times [1,2] and host many socio-economic activities [3]. Coastal areas are very dynamic environments that are subject to modifications even at short timescales. Such modifications depend on several factors, such as climate change, wave action, isostasy, geology, tectonics, and anthropic actions [4]. This is true both for sandy and rocky coasts. Moreover, among the above-mentioned factors, geology and tectonics play a crucial role in the evolution of rocky coasts. In fact, sea cliffs carved in soft sediments evolve more rapidly than sea cliffs carved in hard rock-types [5]. For example, Brooks and Spencer [6] estimated a retreat rate of 0.9–3.5 m/y since 1883 for a sector of the East Anglia sea cliff carved in soft sediments, e.g., mainly unconsolidated sands and clays. Young [7] estimated an average cliff retreat rate of 0.25 m/y since the early 1990s for a sector of the California coastline in the USA, carved in lithified Cenozoic units (e.g., mudstone, shale, sandstone, and siltstone). Epifano et al. [8] estimated an average retreat rate of 0.044 m/y in a 60-year-long period for a rocky coast carved in Jurassic marls and sandstones in Portugal. Furthermore, retreat rates of rocky coasts are also dependent on the amount and density of discontinuities, e.g., bedding, fractures, and faults, even when they affect shore platforms. Naylor and Stephenson [9] analyzed discontinuities within shore platform in Wales and Australia and derived that

the higher the density of discontinuities the more rapidly the shore platform will erode, with consequences also for sea cliffs behind shore platforms.

In this paper we have carried out a geological and geomorphological analysis of a portion of the Campi Flegrei coastline, in the Gulf of Naples (southern Italy). The Campi Flegrei is a volcanic area located along the Tyrrhenian flank of the Southern Apennines (Figure 1). Morphoevolution of the Campi Flegrei has been characterized by the interaction of volcanism and relative sea-level variation due to climate and bradyseism ([10] and reference therein). As a result, both sandy beaches and rocky coasts are present along the coast of the Campi Flegrei. Regarding rocky coasts, they are all carved in volcanic units and are affected by mainly NE-SW and NW-SE trending faults and fractures [11]. These volcanic units include both consolidated tuffaceous deposits, e.g., the Neapolitan Yellow Tuff, hereinafter NYT, aged 15 ka [12], and unconsolidated pre- and post-NYT pyroclastics.

Several studies have been carried out on specific sectors of the Campi Flegrei coast, e.g., the Monte di Procida sea cliff [13], the Miseno Cape [14], the gulf of Pozzuoli [15], and the Posillipo promontory [16]. These studies highlighted the role of ground motions, e.g., bradyseism, besides rock-types, volcano-tectonics, and climate, in shaping the coastline. Furthermore, researches along the Posillipo promontory have been focused on very specific areas, such as the Nisida island [17], the Coroglio sea cliff [18], and the Marechiaro area [19]. Researches on the Posillipo promontory have also pointed out that this amazing portion of the Campi Flegrei coastline has attracted human frequentation since historical times [17,19]. Evidence of this includes the Roman ruins of Palazzo degli Spiriti, the archaeological area of Pausyllipon and the Roman villae of Villa Diana. By the way, despite these very detailed scale analyses, research along the Posillipo coastline has never been performed along a wider portion. To tackle this issue, we analyzed the westernmost portion of the Posillipo promontory, which extends from the Coroglio sea cliff to the NW, to the La Gaiola islet, to the SE (Figure 1).

The main goal of our paper is to provide robust geological and geomorphological field data that may allow us to characterize this sector of the Campi Flegrei coastline with the aim of identifying the areas more susceptible to cliff retreat. The assessment of cliff retreat susceptibility is not a simple task and researchers can adopt various approaches (heuristic, statistic, deterministic ([20] and references therein)) at different scales (from local to regional) [3,20–26]. Our case study is presented at a local scale and is representative of cliffs in volcanic rocks made up of alternating soft and hard pyroclastic deposits. We adopted a heuristic criterion and analyzed the main predisposing factors occurring in the study area. Our aim is not to propose a new method to assess susceptibility but only to present a case study that may be a useful example and source of data for further studies on the stability of similar volcanic coastal cliffs [13,16,18,27].

## 2. Geological and Geomorphological Setting of the Campi Flegrei

The Campi Flegrei is a volcanic area placed at the border of the Campana Plain, a wide tectonic depression occurring along the Tyrrhenian, inner, flank of the Southern Apennines ([28] and reference therein). The Southern Apennines are a NE-vergent fold and thrust belt formed by the collision of the Eurasian and African plates [29,30]. Tectonic depressions along the inner sector of the mountain belt include large peri-Tyrrhenian grabens formed because of the extensional tectonics due to the opening of the Tyrrhenian back-arc basin [31,32]. Among these grabens, the Campania Plain is the largest one. The sedimentary filling of the Campania plain consists of ~3000 m of marine, transitional, and continental deposits, with abundant volcaniclastic deposits produced by both Vesuvius and the Campi Flegrei [28,33,34]. The study area falls within the Campi Flegrei (Figure 1), which is a volcanic field placed in a resurgent caldera (Figure 2).

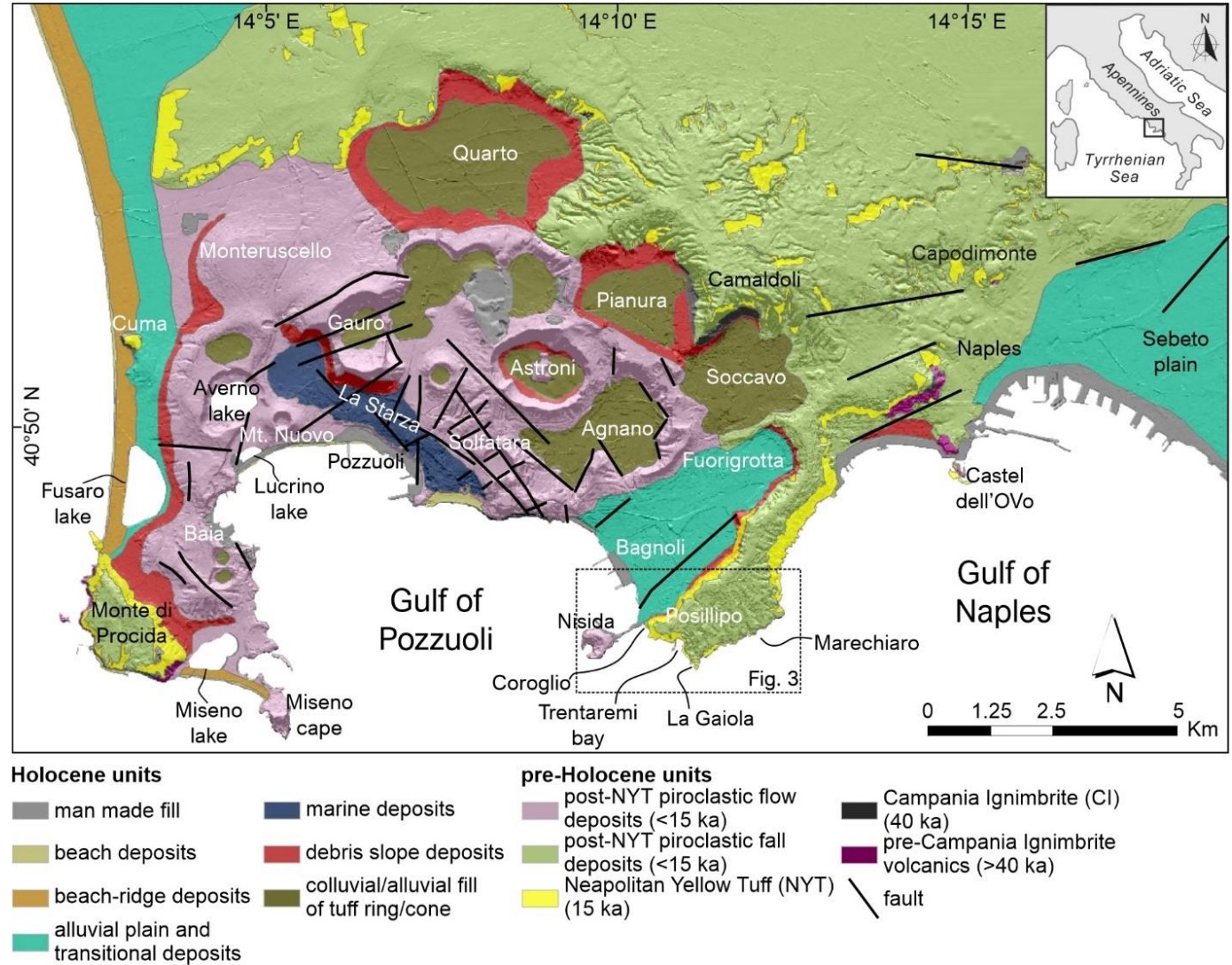

**Figure 1.** Geolithological map of the Campi Flegrei (modified from [10]). Faults are derived from the Geological Sheet 447—Napoli of the Geological Map of Italy at scale 1:50,000, CARG Project [35]. Dashed rectangle indicates location of the Coroglio-Trentaremi sea cliff shown in Figure 3.

The time at which volcanic activity in the Campi Flegrei began is unknown and the oldest outcropping volcanic units are ~60 ka old [36]. The caldera, which is quasi-circular and whose diameter is ~8 km long (Figure 2), formed because of collapses during the two strongest eruptions, which are related to the Campanian Ignimbrite (hereinafter CI, ~39 ka old [37–39]) and the Neapolitan Yellow Tuff (hereinafter NYT, ~15 ka old [12]). The CI eruption emplaced ~300 km$^3$ of pyroclastic fall and flow deposits [40], which have been covered by younger volcanoclastic units. The CI outcrops only in a few sectors of the Campi Flegrei, e.g., at the base of the Camaldoli hill, at Cuma and at the base of the Monte di Procida hill (Figure 1). The NYT eruption extruded at least 40 km$^3$ of pyroclastic fall and flow deposits that accumulated at the caldera boundaries, thus forming the slopes that limit the entire Campi Flegrei (Figure 1) [10]. Moreover, the hills bounding the Campi Flegrei caldera are mainly formed by the NYT; the thickness of these may exceed several tens of meters, they are blanketed by younger pyroclastic fall deposits, and their age is constrained at 15 ka [35]. Volcanic activity after the NYT eruption has been confined within the caldera and consisted of hydromagmatic phenomena with occasional Plinian and effusive events (which are evidenced by the diffuse presence of post-NYT fall deposits and by local outcrops of lavas). Consequently, several monogenic vents formed, including tuff

rings, tuff cones, cinder cones and lava domes (Figure 2) [41–45]. The last volcanic event in the Campi Flegrei occurred in 1538 AD with the formation of Monte Nuovo volcano [46,47] (Figures 1 and 2).

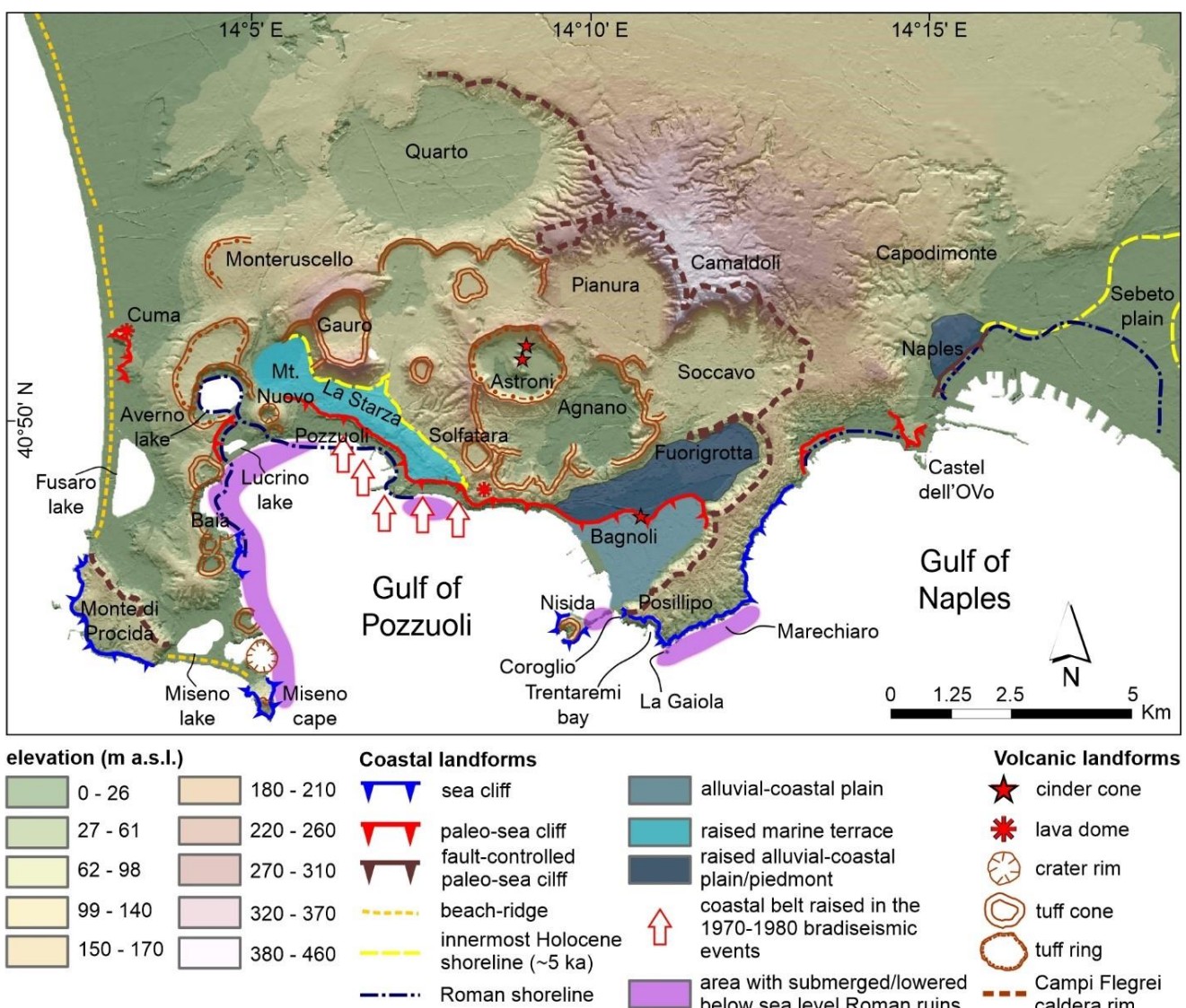

**Figure 2.** Geomorphological map of the Campi Flegrei (modified from [10]).

The Campi Flegrei's history is also characterized, besides volcanic activity, by ground motions in the form of episodes of uplift and subsidence during bradyseismic crises [48,49]. These episodes range from several meters to several tens of meters and continue nowadays (Figure 2) [50–53]. Ground motions are testified to, above all, by the raised marine terrace of La Starza (Figure 2), near Pozzuoli [43,50,51,54], and by submerged archaeological ruins near Baia, Nisida, La Gaiola and Castel dell'Ovo [19,51,52,55,56] (Figure 2).

The complex volcanic history of the Campi Flegrei has influenced its geomorphological setting, which results in a hilly landscape mainly modelled by slope, volcanic and coastal processes. Elevation within the Campi Flegrei spans from 0 m a.s.l. to 460 m a.s.l., with the highest peak corresponding with the Camaldoli hill (Figure 2). Slope processes include landslides affecting the volcanic slopes that often cause severe damage to people and buildings [57]. Volcanic landforms are clustered within the caldera's inner slope and consist of tuff rings, tuff cones, cinder cones and lava domes formed by volcanic events in the last 15 ka ([10] and reference therein).

Coastal landforms consist of both sandy and rocky coasts. Sandy coasts occur near the main alluvial plains, e.g., the Sebeto and the Fuorigrotta-Bagnoli plain (Figure 2) and suffered man-made intervention and strong urbanization. Sandy beaches are also present to the west of the Campi Flegrei where they still preserve remnants of beach-ridge and pass, towards the inner side, to coastal lakes (e.g., Fusaro lake, Miseno lake and Lucrino lake; Figure 2). Rocky coasts form high and steep sea cliffs occurring near Miseno Cape and around Posillipo. Sea cliffs are cut in the pre-NYT pyroclastics, in the NYT and in the post-NYT pyroclastics. In addition, sea cliffs are dissected by a dense network of faults and fractures, mainly NE-SW and NW-SE trending [11], which may act as predisposing factors for rock falls. Sea cliff instability is also enhanced by diffuse extraction of tuff deposits, since Roman times, which results in anthropic cavities at the base of most of the sea cliffs [10]. Rocky coasts include our study area, which is a sector of the Posillipo coastline, between the Coroglio sea cliff, to the west, and the Trentaremi bay and La Gaiola islet, to the east (Figures 2 and 3).

## 3. Materials and Methods

The study was based on the geological and geomorphological analysis of the Coroglio-Trentaremi sea cliff by combining field work with analysis of detailed scale topographic maps and orthophotos, and stratigraphical data from deep boreholes.

Geomorphological analysis has been carried out by means of a detailed scale topographic map (Technical Map of the Regione Campania at scale 1:5000), Google Earth images and field work.

This analysis allowed us to reconstruct the main geomorphological features of the sea cliff, and to map the main landform indicators of slope instability. We gave particular attention to detachment niches and landslide bodies and to the presence of notches and cavities at the base of the sea cliff. The latter are indicators of undercutting by waves, which is undoubtedly the most important factor in causing coastal retreat [4,58]. The rate of undercutting is controlled by the complex and wide-ranging behavior of geological materials and by the great variability in their geotechnical properties. According to Budetta [24], the increasing depth of the notch into fractured rock mass, because of wave erosion, is responsible for the spreading of the shear stresses towards the top of the cliff and for the destabilization of the overlying slope. We also considered the presence of ancient quarries (mainly roman in age) for tuff extraction which, accordingly to Ruberti et al. [27], may also be considered as instability factors, especially if involved in the shear stress propagation caused by waves undercutting.

Geological analysis was conducted by means of field work, both inland and by the sea using small boats that allowed to reach some areas not accessible by foot (e.g., most of the sea cliff base), and borehole data [59]. The combination of field and borehole data allowed us to derive 6 geological cross-sections representative of the subsurface stratigraphical setting of the Coroglio-Trentaremi sea cliff. Furthermore, field work also allowed us to define the structural setting of some portions of the sea cliff by establishing 7 structural stations. Structural stations have been established directly on the cliff by climber geologists along 10 vertical rope descents [59]. These structural data have been combined with field data collected at the base of the sea cliffs. As a result, we have been able to establish the relationships between bedding, sea cliff trend, and fractures by defining dip and dip direction and to propose failure mechanisms for each structural station.

By comparing geological and geomorphological collected data, we used a heuristic approach to identify and select two main groups of predisposing factors which are listed in Table 1. The first group includes factors that may influence landslide intensity (e.g., sea cliff height, I1; volume of landslide body, I2; volume of detachment niches, I3; volume of blocks or projecting sectors, I4), and factors that may increase the weakness of a sea cliff (e.g., fracture spacing, W1; persistence of beating fractures, W2; volume of caves at the sea cliff base, W3; distance between the shoreline and the base of active sea cliff, W4). The volume of the caves has been derived by expeditive surveys with a laser pointer. The distance

between the present shoreline and the base off the cliff considers the presence/absence of a debris body that may act as protection from the wave action. Each factor includes four group values, which have been weighted according to their attitude to instability.

**Table 1.** Instability (I) and weakness (W) factors adopted to evaluate the geomorphological zonation to landslide along a sea cliff, with relative weight.

| Weight | I1 Sea Cliff Height (m) | I2 Volume of Landslide Body (m³) | I3 Volume of Detachment Niche (m³) | I4 Volume of Blocks or Projecting Sectors (m³) | W1 Fracture Spacing (m) | W2 Persistence of Beating Fractures (m) | W3 Volume of Caves at the Sea Cliff Base (m³) | W4 Distance between the Shoreline and the Base of Active Sea Cliff (m) |
|---|---|---|---|---|---|---|---|---|
| 1 | 0–20 | 0–10 | 0–100 | 0–5 | >20 | 0–5 | 0 | >20 |
| 2 | 21–100 | 11–30 | 101–300 | 6–10 | 10–19 | 6–10 | 1–20 | 11–20 |
| 3 | 100–200 | 31–50 | 301–1000 | 11–30 | 5–9 | 11–30 | 21–100 | 2–10 |
| 4 | >200 | >50 | >1000 | >30 | 0–4 | >30 | 101–1000 | 0–1 |

To evaluate the susceptibility to landslide of a sea cliff, we have crossed the weights of the Intensity (I) and the Weakness (W) factors (Table 2). This matrix suggests that high magnitude (M) landslides may occur when high weight I and W factors occur. As an example, either a very high sea cliff or a sea cliff with a high volume of blocks or projecting sectors coupled with large caves at the sea cliff base may contribute to high magnitude landslides (M3 class in Table 2). On the other hand, a low height sea cliff carved in poorly fractured rocks and with no caves at the sea cliff base may contribute to low magnitude landslides (M1 class in Table 2).

**Table 2.** Matrix for the evaluation of landslide magnitude (M) by crossing weights of both the Intensity (I) and Weakness (W) factors. Red, orange and yellow colors indicate high (M3), moderate (M2) and low (M1) magnitude classes.

| Weight of Weakness Factor | Weight of Intensity Factor | | | |
|---|---|---|---|---|
| | 1 | 2 | 3 | 4 |
| 1 | M3 | M3 | M2 | M2 |
| 2 | M3 | M3 | M2 | M1 |
| 3 | M2 | M2 | M2 | M1 |
| 4 | M1 | M1 | M1 | M1 |

## 4. Results

### 4.1. Geological Setting of the Coroglio-Trentaremi Sea Cliff

The geological setting of the Coroglio-Trentaremi sea cliff is reported in Figure 3, whereas the stratigraphical relationships between the outcropping rock-types are reported in the stratigraphical columns of Figure 4 and in the geological cross-sections of Figure 5. The sea cliff is mainly cut in tuffaceous deposits of the NYT and in older volcanics, that are mantled by unconsolidated post-NYT pyroclastics.

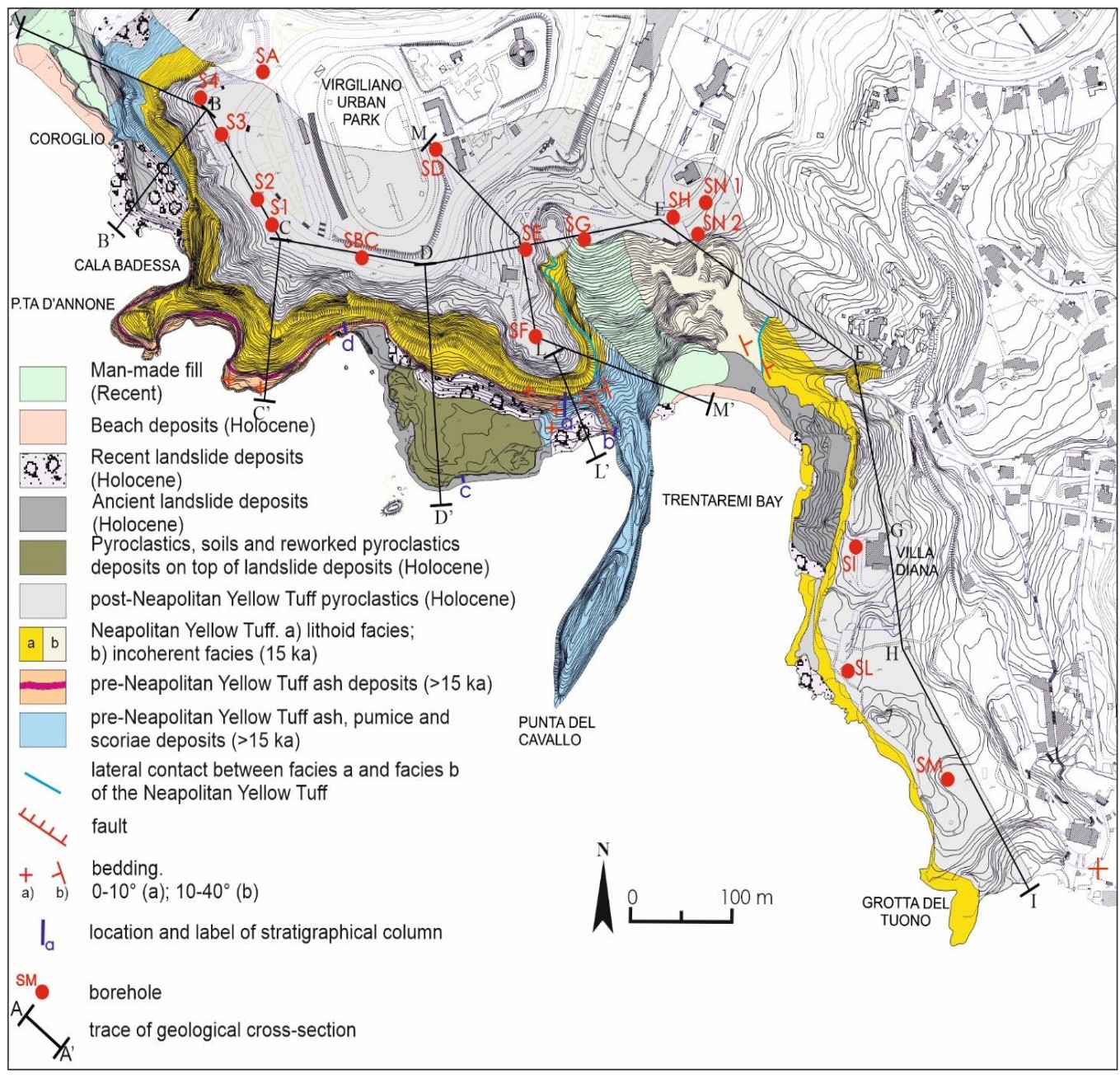

**Figure 3.** Geological map of the Coroglio-Trentaremi area.

The oldest volcanic unit consists of whitish and stratified tuff deposits, made of a pre-NYT ash, pumice and scoriae (Figure 3), which outcrops in the Coroglio area, to the west, and at Punta Cavallo and Trentaremi bay, to the east (stratigraphical columns A and B in Figure 4). Geological cross-sections A-A', B-B', L-L' and M-M' in Figure 5 show that this unit forms a morphostructural low where the more recent pre-NYT ashy unit and the NYT accumulated. This unit is strongly inclined in the surroundings of the Trentaremi bay (dip around 45°) and its bottom is made of boulders, scoriae and large pumices, suggesting that this area is pretty close to the possible source area, as has been already hypothesized by Cole et al. [60]. The unit continues upward with greyish-whitish, pisolithic ashes and pumices, with interbedded large pumice-rich layers, whose bedding ranges from plan-parallel to gently dipping and slightly undulating. The base of this unit is not recognizable, whereas its top surface has a probably erosional origin. The age of this unit ranges between 37 ka and 15 ka and is constrained at 20.9 ka by Scarpati et al. [61] (Figure 3).

The pre-NYT ash, pumice and scoriae deposits pass upward to a mainly ashy unit, which has been named pre-NYT ash deposits in Figure 3. This unit outcrops to the west of the study area, near Punta d'Annone (geological cross-sections A-A', C-C' and D-D' of Figure 5) and its thickness ranges between 20 and 30 m (stratigraphical column D in Figure 4). It consists of an alternation of horizontally stratified ashy layers and undulated scoriae rich and ashy layers probably of fall origin. This unit is either not reported in the literature or it is grouped with the NYT [35], its age ranges between 37 ka and 15 ka, and it is tentatively correlable with the Whitish Tuff of Scarpati et al. [62], which is dated at 19.7 ka (Figure 3).

The pre-NYT ash deposits pass upward to the NYT, which outcrops in the entire study area, with thickness ranging from 30 m (to the east, near the Trentaremi bay) to 80–100 m (to the west, near the Coroglio sea cliff) (Figures 3 and 5). It consists of lithified pyroclastic deposits made of pumices, lithics and cm-sized scoriae, yellowish to greenish, in a yellowish ashy matrix [63]. The unit is massive (as a typical pyroclastic flow deposit) being just locally thinly layered and with abundant sub-vertical fractures. The top surface of the NYT is a low-dipping erosional surface and its age is 15 ka [12].

The NYT passes upward to fall deposits related to Phlegrean eruption which occurred in the last 15 ka [43,60], which have been named post-NYT pyroclastics in Figure 3. This unit consists of an alternation of decimeter thick sandy and ashy layers and centimeter thick layers rich in whitish pumices with interbedded thin paleosols that testify to a period of quiescence of the eruptive phases. A paleosol also occurs between the top surface of the NYT and the base surface of the post-NYT pyroclastics. The top surface of the post-NYT pyroclastics is almost flat and corresponds with the topographic surface (Figures 3 and 5). The thickness of this unit is not uniform (Figure 5), ranging from 10–15 m (to the east, near the Trentaremi bay) to 40 m (to the west, near the Coroglio sea cliff), and its age is younger than 15 ka (Figure 3).

The sea cliff foothill, from the Coroglio area to the Trentaremi bay, is mantled by slope debris, landslide bodies and beach deposits, which are mainly located near the Trentaremi bay and whose maximum thickness is around 20 m (Figure 3 and stratigraphical column C in Figure 4). Moreover, landslide deposits consist of reworked ashes with NYT blocks, whose size may exceed some cubic meters, and are due to rock falls affecting the sea cliff, some of which are very recent as suggested by the lack of vegetation and the low erosion by sea waves.

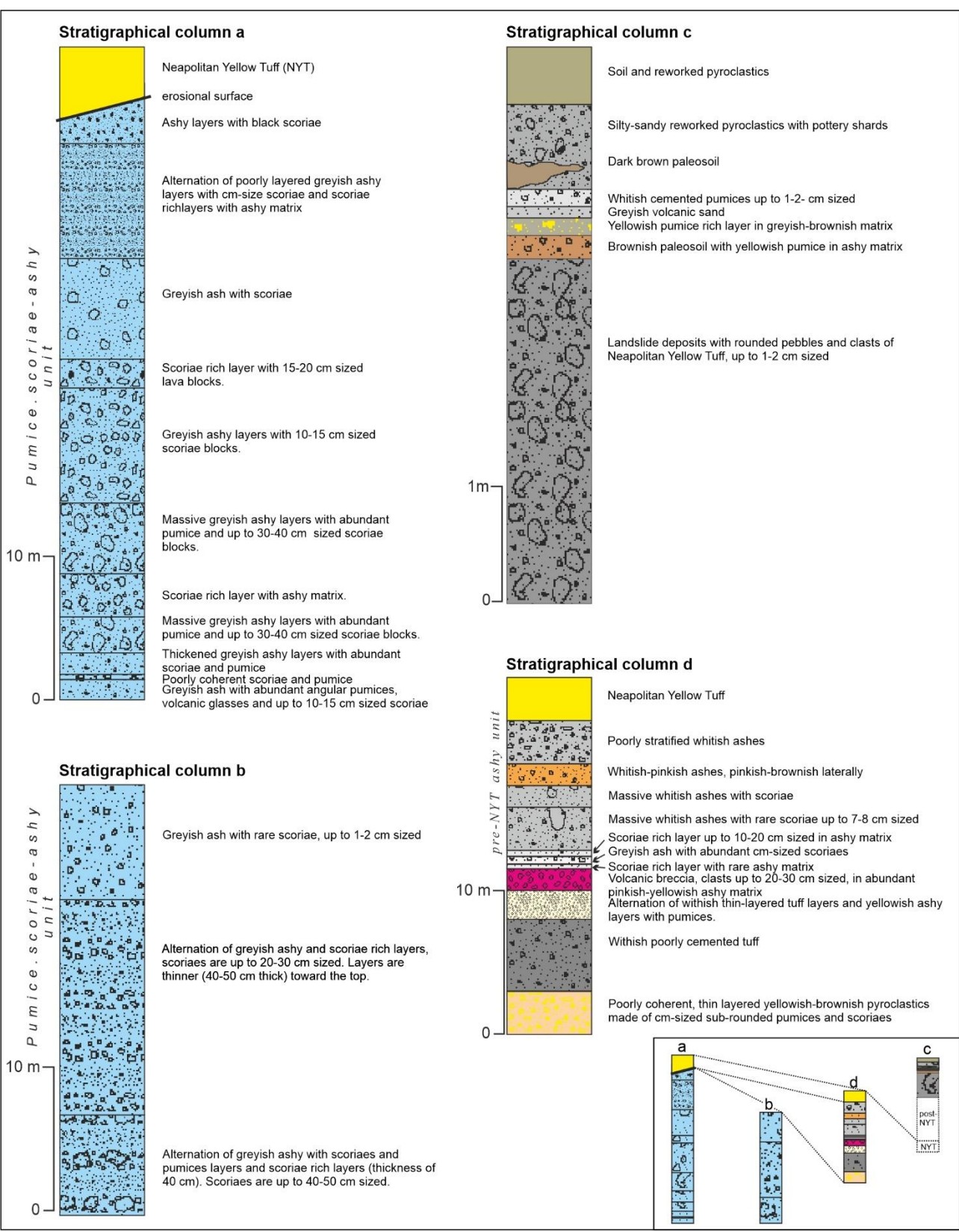

**Figure 4.** Stratigraphical columns representative of the geological setting of the Coroglio-Trentaremi sea cliff base. Location of stratigraphical column is reported in Figure 3. Inset scheme in the lower right corner of the figure shows stratigraphical correlation between stratigraphical columns.

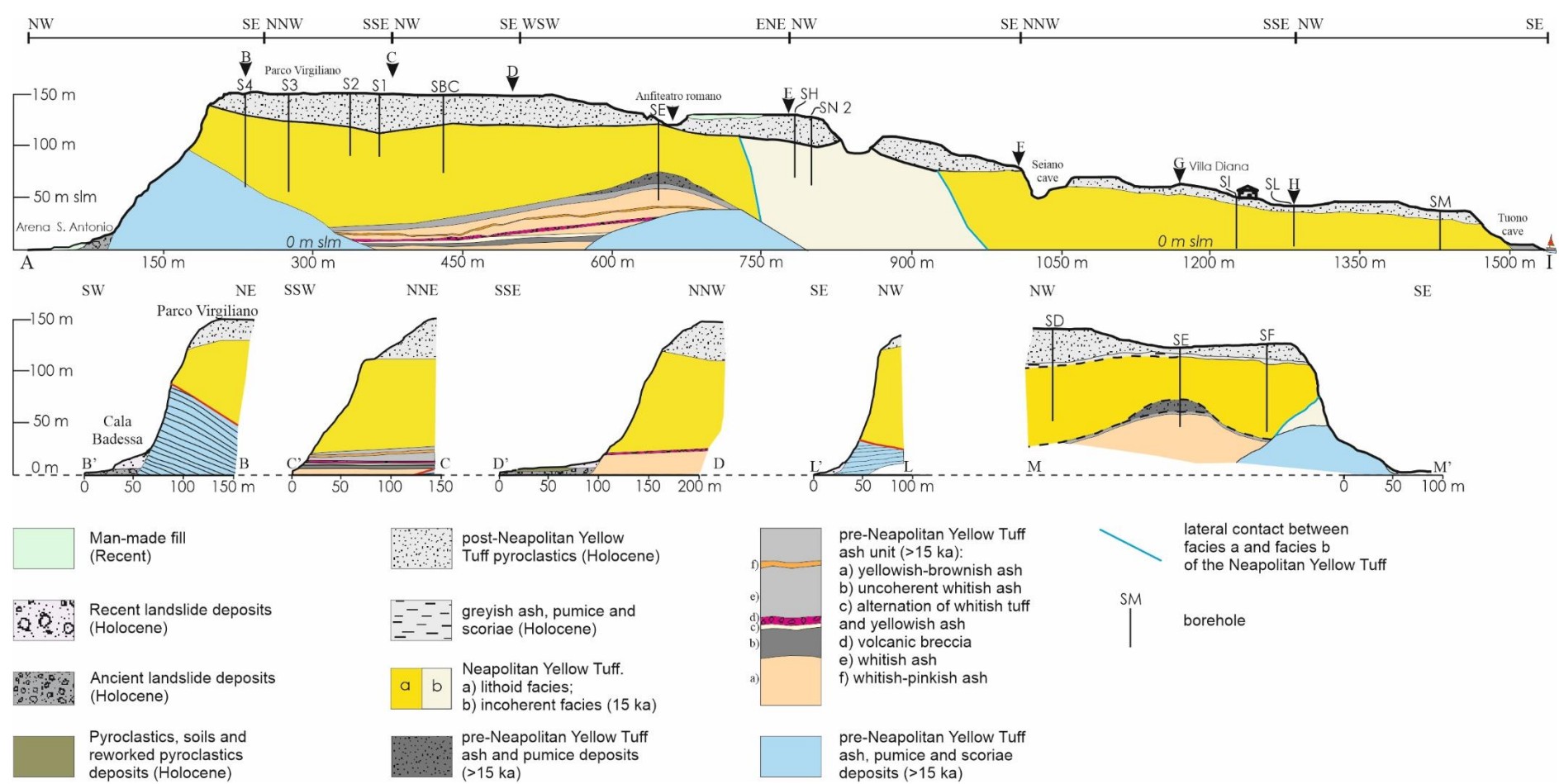

**Figure 5.** Geological cross-section along the Coroglio-Trentaremi sea-cliff. Cross-section location is reported in Figure 3.

### 4.2. Geomorphological Setting of the Coroglio-Trentaremi Sea Cliff

Geomorphological analysis has been addressed to the recognition of the predisposing factors that may cause landslides and cliff retreat, and the results are reported in the geomorphological map of Figure 6. The study area is characterized by high sea cliffs that are prone to retreat because of both their stratigraphical setting (alternation of poorly cemented pyroclastics and tuffs) and the occurrence of a dense net of sub-vertical fractures in the tuffaceous deposits.

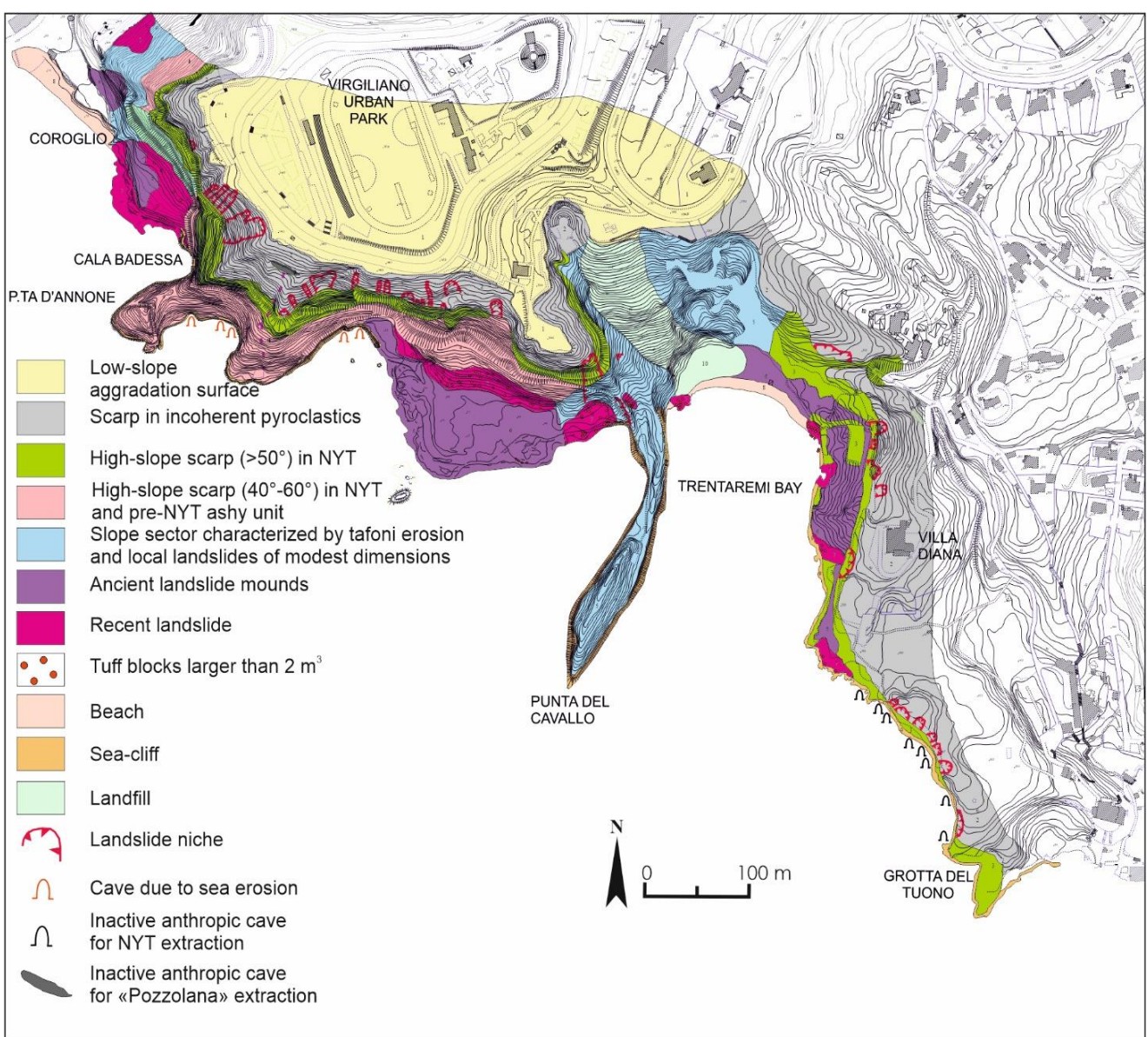

**Figure 6.** Geomorphological map of the Coroglio-Trentaremi area.

The geomorphological map of Figure 6 highlights the diffuse presence of landslides, which have also been reported on Google Earth 3D view (Figure 7), where they have been classified according to the type of movement and the type of material involved. Moreover, flow-like landslides (yellow dots in Figure 7) affect the sectors of the Coroglio-Trentaremi sea cliff where post-NYT pyroclastics outcrop and with slope angle between 45° and 70°, whereas rock-falls (red dots in Figure 7) occur in the NYT with slope angle between 70° and 90°. We also mapped in Figure 7 the caves at the base of the sea cliff, which are

mainly due to anthropic activities for the NYT extraction. The collected data clearly suggest that the main factors influencing the types of landslide are lithology and slope degree (Figure 8). Flow-like landslides remain confined to the upper part of the cliff, where soft unconsolidated deposits rest on top of the NYT formation. The presence of hard/soft rock-types contacts along the sea cliff may also increase the possibility of detachment of rock falls because of selective erosion. Also, the presence of caves (both natural and artificial) plays a crucial role in the evolution of the sea cliff by identifying sectors more prone to instability because of the presence of vacuums that may enhance sea erosion at the base of the cliffs.

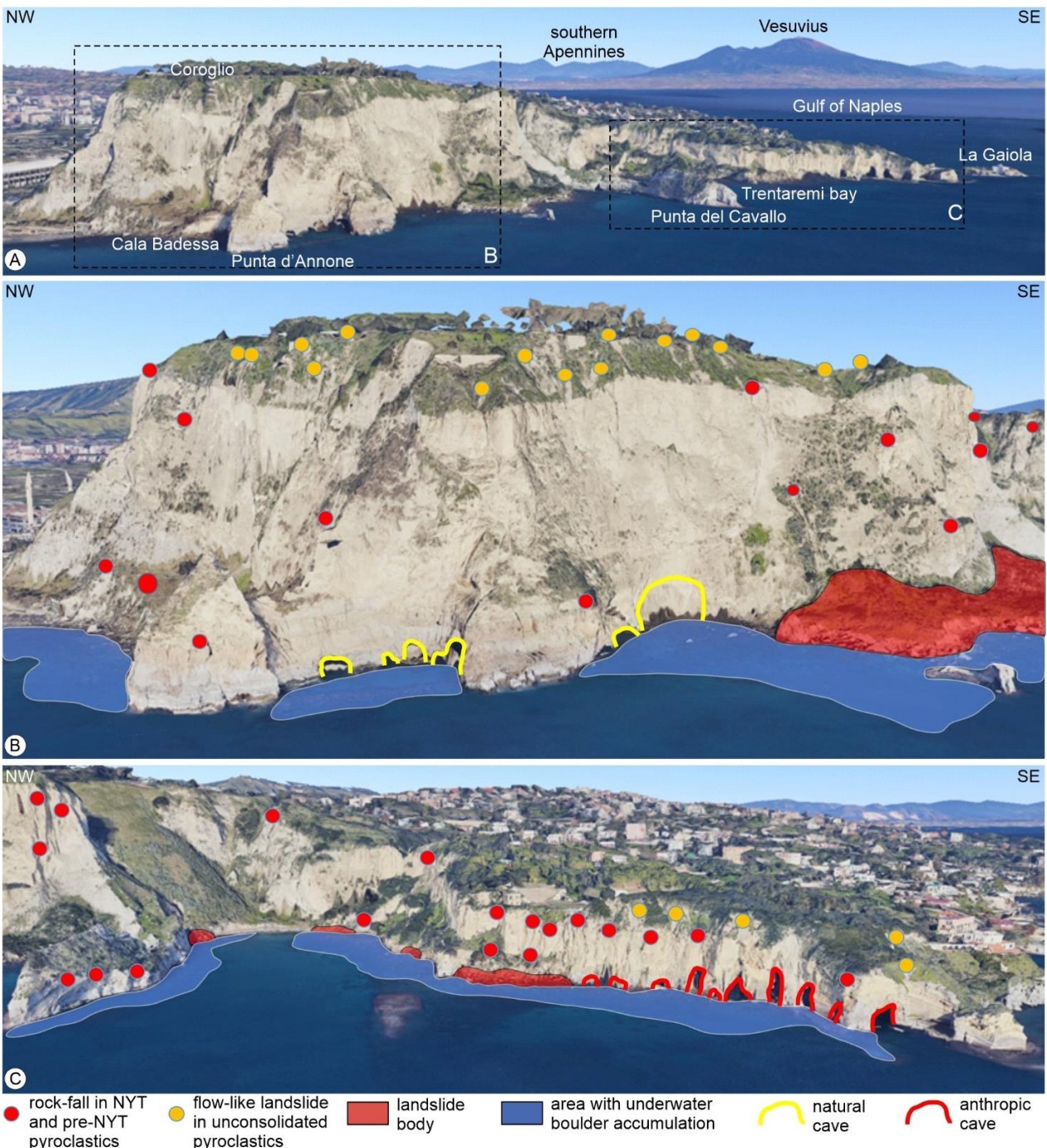

**Figure 7.** (**A**) Google Earth 3D views of the Coroglio-Trentaremi coastal sector. (**B**) Detail of the Coroglio sea cliff. (**C**) Detail of the Trentaremi sea cliff.

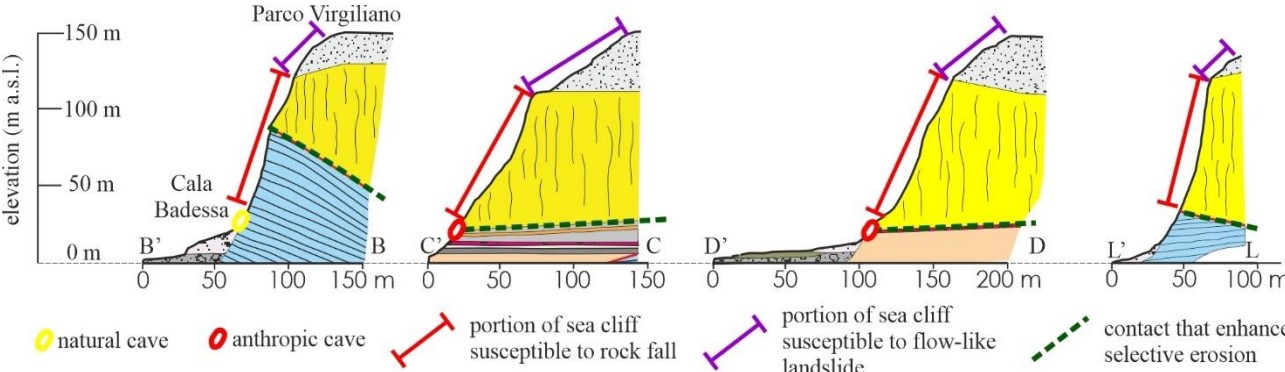

**Figure 8.** Main predisposing factors controlling cliff stability along the Posillipo promontory. Geological cross-sections are modified from Figure 5. Vertical lines in the NYT indicate sub-vertical fractures affecting this unit.

In Figure 9 we show some photos of the sea cliff that allow us to appreciate the size of the landslide bodies, which are up to some hundreds of cubic meters. Landslide bodies of different ages may also be appreciated because of the lack of vegetation, which testify for the high frequency of rock falls along this sea cliff. The tuffaceous deposits of the NYT are also locally affected by a very dense fractures network that shapes the sea cliff profile. Fractures are in many cases open, thus suggesting the possible occurrence of a landslide event in the near future (Figure 10).

### 4.3. Morphostructural Analysis and Failure Mechanism

Faults and fractures trends have been analyzed by combining a morphostructural analysis of Google Earth orthophotos and detailed scale topographic maps with field surveys aimed at collecting structural data in seven geo-mechanical stations.

Morphostructural lineaments along the Posillipo coast mainly trend N20, N45 and N135 (Figure 11A). Moreover, the more diffuse trends are the N20 and N45, which result to be parallel to the fault that bound the Posillipo promontory towards the NW, i.e., towards the Bagnoli-Fuorigrotta plain [35] (Figure 1).

In addition, NW-SE trending faults and fractures have also been recognized in most of the structural stations along the Posillipo sea cliff, in the area between the Coroglio promontory and the Trentaremi bay (structural stations A to E in Figure 11B). The only exceptions are the structural stations F and G, near the Trentaremi bay, which also record some N-S trend. Faults and fractures measured in all the structural stations have high dip angles that range between 75° and 90°.

The diffuse presence of both NE-SW and NW-SE trending faults and fractures is consistent with data from Vitale and Isaia [11], who performed detailed structural analysis in the Campi Flegrei caldera, and suggested that these trends are the prevalent ones.

Overall data from morphostructural analysis and from the structural stations point out the occurrence of a pervasive system of sub-vertical, mainly NE-SW and NW-SE trending faults and fractures that affect the tuff deposits. These systems also cause the formation of rupture surfaces from which rock-falls originate, thus conditioning the stability of the Coroglio-Trentaremi sea cliff.

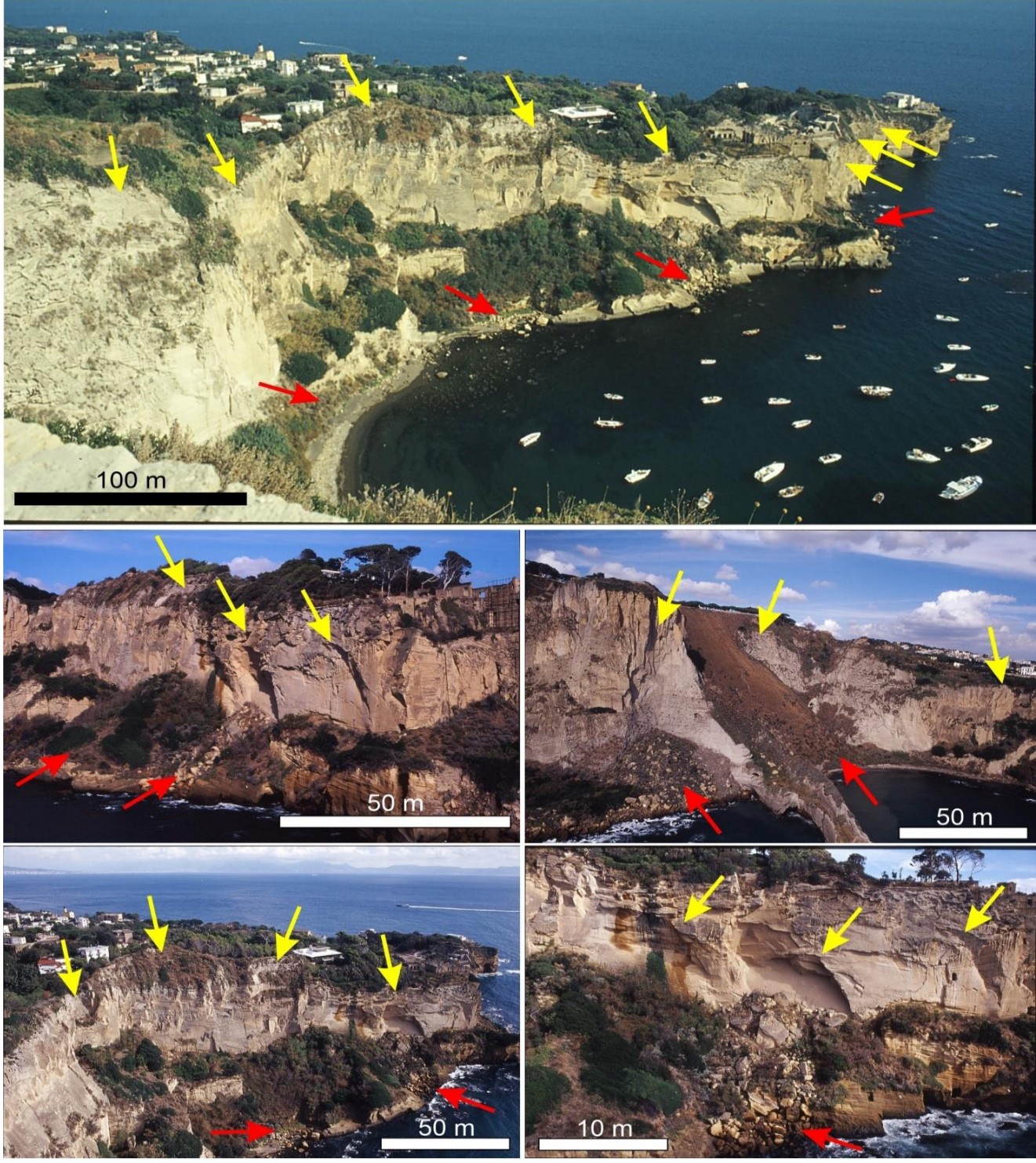

**Figure 9.** Landsliding along the Coroglio-Trentaremi sea cliff. Red arrows indicate landslide bodies, whereas yellow arrows indicate detachment niches.

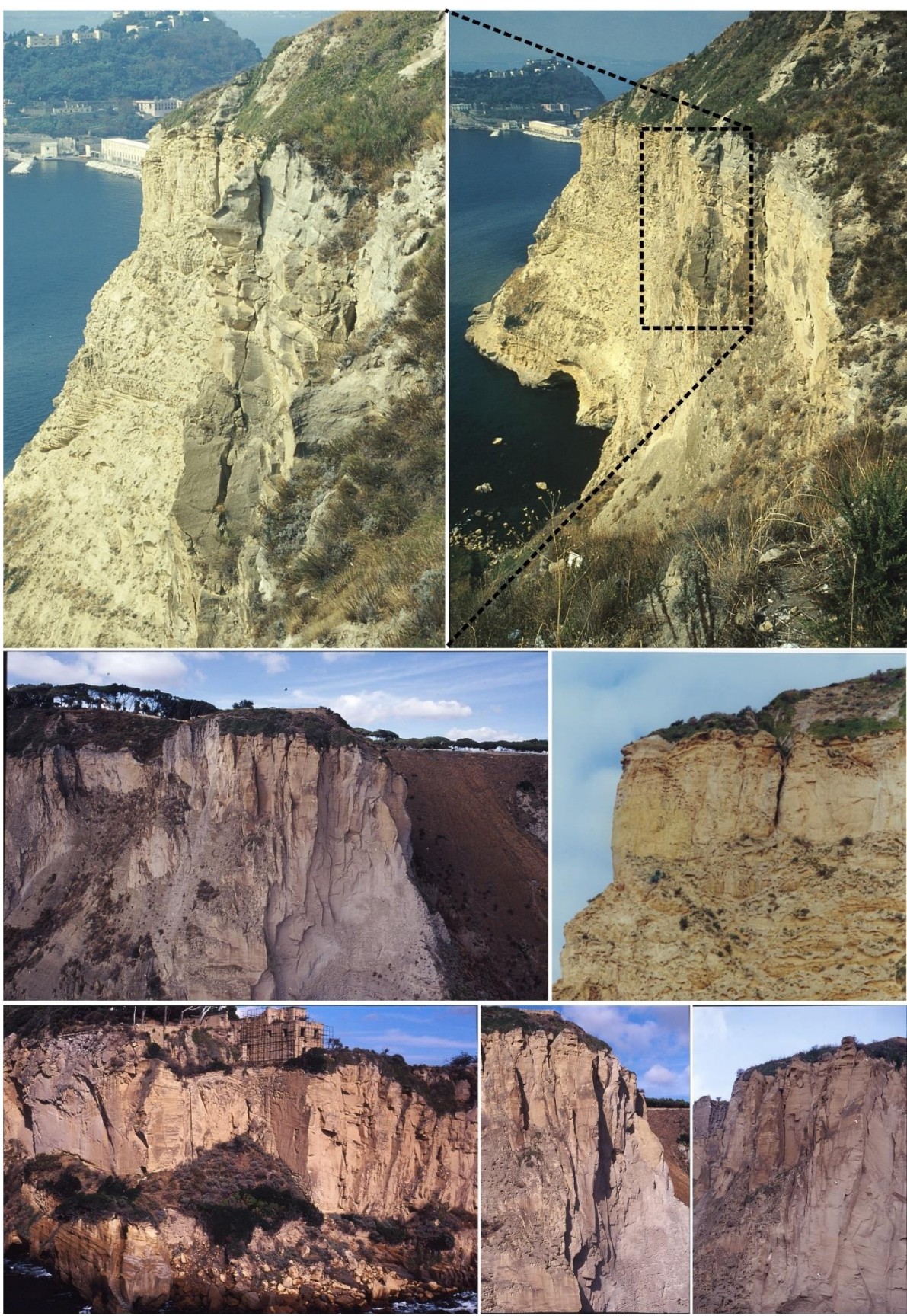

**Figure 10.** Beating fractures and rock-falls niches along the Coroglio-Trentaremi sea cliff. Unstable fractures, dihedrons and pinnacles set on high angle systems are well evident.

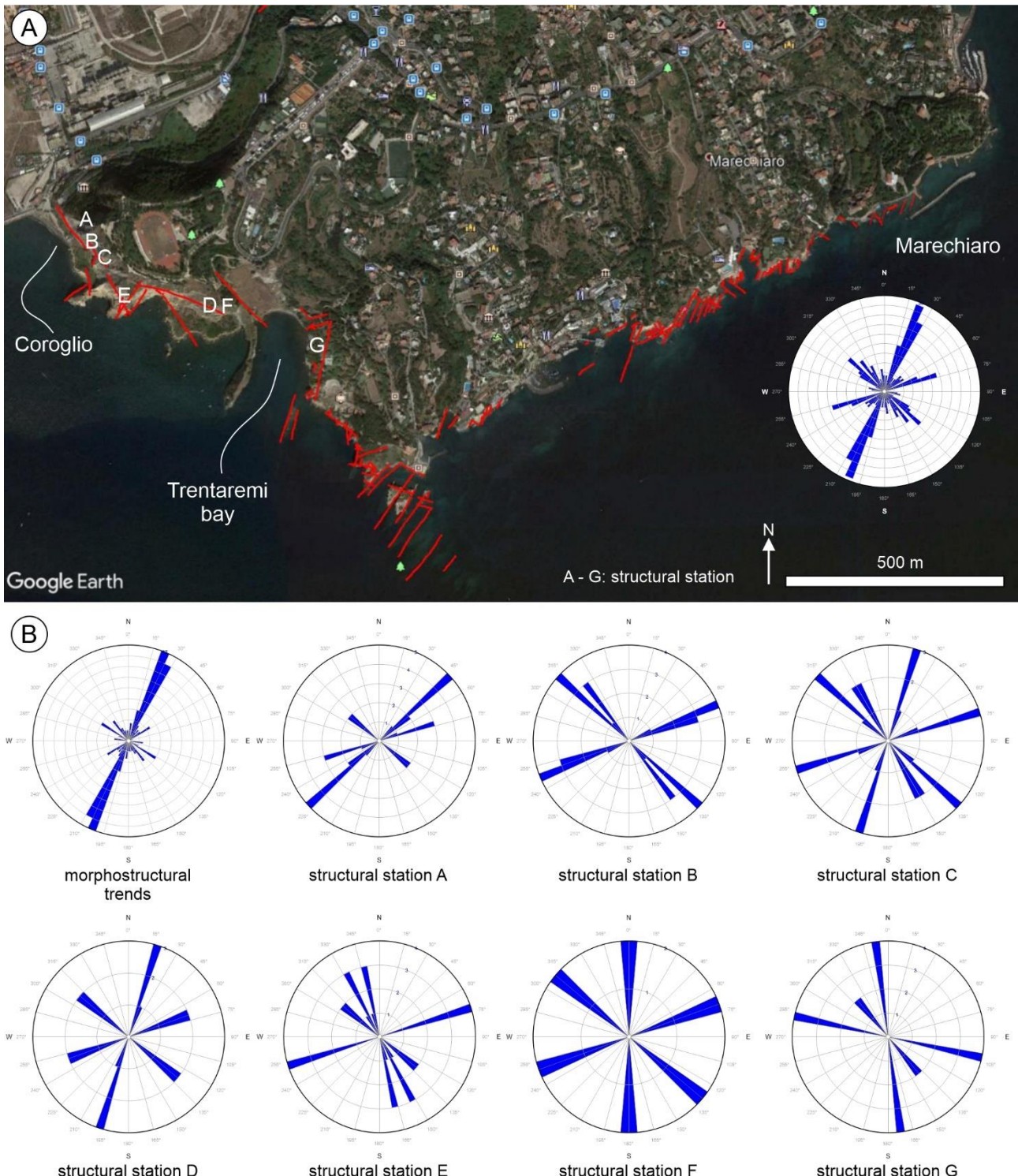

**Figure 11.** (**A**) Morphostructural lineaments (red lines) along the Coroglio-Trentaremi sea cliff plotted on Google Earth orthophoto. Labels A to G indicate the location of the structural stations. The rose diagram to the right of panel A shows the direction of all the morphostructural lineaments, the faults and fractures shown in the rose diagrams of panel B. (**B**) Rose diagrams showing the direction of the morphostructural lineaments and the direction of the faults and fractures measured in the single structural stations along the Coroglio-Trentaremi sea cliff. Roses represent the absolute number of data.

To define the failure mechanism of the tuffaceous cliff we analyzed field data from seven structural stations, whose locations are shown in Figure 11. All the structural stations highlight the occurrence of a dense net of fractures that, in many cases, allow us to identify

several low-equilibrium sectors in the sea cliff. More specifically, we recognized five systems of fractures and bedding whose dip-direction and dip are listed in Table 3.

**Table 3.** List of fracture systems (K1 to K5) along the Coroglio-Trentaremi sea cliff with indication of the dip-direction (α) and dip (β). Bedding (Ks) and sea cliff front trend are also reported. Structural stations' locations are reported in Figure 11.

| Structural Station A | α | β | Structural Station B | α | β |
|---|---|---|---|---|---|
| K1 | 310 | 85 | K1 | 310 | 85 |
| K2 | 250 | 85 | K2 | 250 | 85 |
| K3 | 45 | 85 | K4 | 140 | 70 |
| Ks | 200 | 20 | Ks | 200 | 20 |
| sea cliff front | 230 | 85 | sea cliff front | 245 | 85 |
| **Structural Station C** | **α** | **β** | **Structural Station D** | **α** | **β** |
| K1 | 310 | 85 | K1 | 310 | 85 |
| K2 | 250 | 85 | K2 | 250 | 85 |
| K5 | 330 | 70 | Ks | 200 | 20 |
| Ks | 330 | 70 | sea cliff front | 200 | 80 |
| sea cliff front | 200 | 20 | | | |
| **Structural Station E** | **α** | **β** | **Structural Station F** | **α** | **β** |
| K1 | 310 | 85 | K1 | 310 | 85 |
| K2 | 250 | 85 | K2 | 250 | 85 |
| K5 | 330 | 70 | Ks | 200 | 20 |
| Ks | 200 | 20 | sea cliff front | 180 | 85 |
| sea cliff front | 165 | 80 | | | |
| **Structural Station G** | **α** | **β** | | | |
| K4 | 140 | 70 | | | |
| K5 | 175 | 65 | | | |
| Ks | 200 | 20 | | | |
| sea cliff front | 280 | 85 | | | |

The intersection between fracture systems and bedding allows us to hypothesize different failure mechanisms along the Coroglio-Trentaremi sea cliff, which are listed in Table 4. To synthesize, structural analysis of the sea cliff points out a high susceptibility to landsliding of several tuffaceous blocks by means of rock-fall and toppling. Systems of fractures are 5 to 7 m spaced and intercept the bedding, thus isolating large rocky dihedrons and pinnacles whose volume may exceed some hundreds of cubic meters (Figure 10). This volume estimation is confirmed by measurements of landslide niches' dimensions, that may reach values of 500 m$^3$, and of large blocks within the landslide bodies.

**Table 4.** Failure mechanism proposed for each structural station.

| Structural Station | Fractures' System | Proposed Failure Mechanism |
|---|---|---|
| A | K1-K2-K3-Ks | Toppling |
| B | K1-K2-K4-Ks | Toppling |
| C | K1-K2-K5-Ks | Wedge breaks and planar slide |
| D | K1-k2-Ks | Rock-fall |
| E | K1-K2-K5-Ks | Rock-fall and toppling |
| F | K1-K2-Ks | Rock-fall and toppling |
| G | K4-K5-Ks | Wedge break |

*4.4. Geomorphological Zonation of Instability along the Coroglio-Trentaremi Sea Cliff*

Overall geological, structural, and geomorphological data highlight the diffuse presence of landslides affecting the sea cliff. We propose, in Figure 12, a geomorphological zonation of the Coroglio-Trentaremi sea cliff, which highlights the location of areas more susceptible to instability. This geomorphological zonation is based on the evaluation of I

and W factors (Table 1), whose weights have been crossed according to the matrix reported in Table 2.

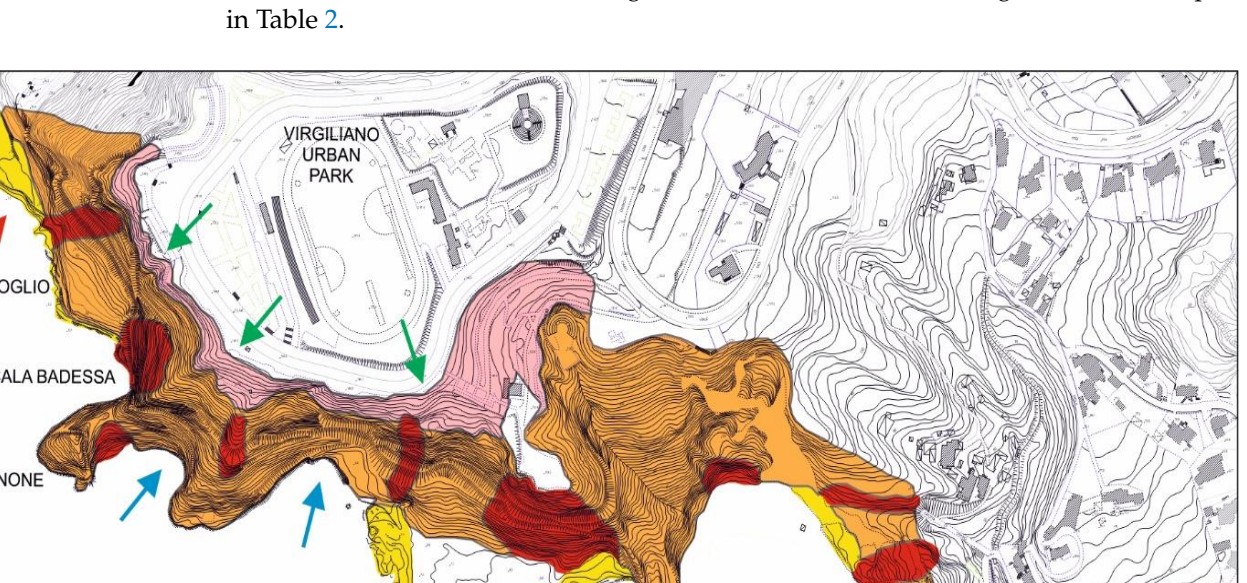

**Figure 12.** Geomorphological zonation of landslide magnitude along the Coroglio-Trentaremi sea cliff. M3: high magnitude landslide (hundreds to thousands of cubic meters involved); M2: moderate magnitude landslide (tens of cubic meters involved); M1: low magnitude landslide (few cubic meters involved). Arrows indicate areas subject to risk due to cliff edge retreat (green arrows), to the presence of sandy shore for mooring boats (red arrows) and to the presence of ships and swimmers (blue arrows).

The map shows that the more diffuse magnitude class is M2 (e.g., moderate magnitude). High magnitude landslides (class M3 in Figure 12) are limited to few sectors of the Coroglio-Trentaremi sea cliff. These include portions of the sea cliff with high sea cliff height values, presence of a dense network of fractures, some of which are beating fractures, and with large caves at the base of the sea cliff. Low magnitude landslides (class M1 in Figure 12) include few sectors at the base of the sea cliff, occurring near Coroglio, the Trentaremi bay and the La Gaiola islet. In this area, the sea cliff height is low to very low and landslide bodies locally protect the sea cliff base from wave action.

Sectors of the sea cliff where the presence of human activities may enhance landslide risk are limited to few portions of the Coroglio-Trentaremi sea cliff. These include both areas placed on top of the sea cliff edge and areas placed at the base of the sea cliff. On top of the sea cliff, the area with sport facilities at the Virgiliano urban park, to the west, and

the archaeological roman villae of Diana, to the east, are present (Figure 12). In both cases, cliff edge retreat due to landslide may cause several problems to both human activities and archaeological finds. At the base of the sea cliff, sandy shores for mooring boats are present, which may increase landslide risk especially in summer due to the increasing number of boats that attend this sector of the Campi Flegrei coastline.

## 5. Discussion and Conclusions

Overall data highlight the complex geological, geomorphological and morphostructural setting of the Coroglio-Trentaremi sea cliff, which is carved in volcanic deposits whose ages range between 20 ka and the present. The main volcanic units are made up of both loose (mainly ashy and sandy) and lithified deposits (generally tuffs). The contact between the different volcanic units is characterized by volcano-tectonic collapses and erosional surfaces locally marked by paleosols, which also control the bedding of strata. All these conditions represent significant predisposing factors for selective erosion and, consequently, for cliff instability.

The lithified deposits are affected by a pervasive system of sub-vertical faults and fractures, mainly NE-SW and NW-SE trending, which may favor the occurrence of landslides. We recognized five systems of fractures, a finding that is consistent with geostructural analysis along the Coroglio cliff by Matano et al. [16]. The interaction between bedding and fracture' systems determine the failure mechanism, which mainly consists of rock fall and toppling that may involve up to some hundreds or even thousands of cubic meters of material.

Landslide type and related volumes are consistent with detailed topographic data produced by Caputo et al. [64] along the Coroglio sea cliff, who recognized rock fall, debris fall, earth flow and soil slip that caused, in the 2013–2015 time span, the accumulation of 210 m$^3$ of material. Furthermore, Caputo et al. [64] also estimated average cliff retreat in the 2013–2015 time span to be of 0.07 m/yr.

The cliff zonation that we propose in this study is based on an empiric geomorphologic approach and considers a high quantity of field data. We know that it is not exhaustive and that to better define the landslide risk, further investigations are necessary. Nonetheless, it may enable the development of a reliable scenario of cliff instability, which may be useful either to plan intervention works in the most critical areas or to organize prevention plans aimed at risk mitigation.

The studied coastal segments surely have high naturalistic and historical value, including the beautiful bays of Coroglio, Trentaremi and, at the top of promontory, the Pausyllipon- La Gaiola archaeological site and the Virgiliano urban park.

Our study pointed out that, at some points, landslide niches are quite close either to some important archaeological remains, such as the Roman villae of Diana near the Trentaremi bay, or to touristic areas, such as the Virgiliano Urban Park on top of the Coroglio cliff, also showing evidence of cliff retreat (Figure 12). These data suggest that these sectors deserve mitigation intervention to prevent damage to people and to man-made structures, and to preserve this unique and peculiar site for future generations.

**Author Contributions:** Conceptualization, A.S. and G.D.C.; methodology, A.S. and G.D.C.; software, E.V.; validation, A.S. and N.S.; formal analysis, E.V.; investigation, A.S., G.D.C., N.S. and E.V.; writing—original draft preparation, A.S., N.S. and E.V.; writing—review and editing, A.S., N.S. and E.V.; visualization, G.D.C.; supervision, A.S. and N.S. All authors have read and agreed to the published version of the manuscript.

**Funding:** This research received no external funding.

**Acknowledgments:** The authors wish to thank three anonymous reviewers whose suggestions helped us to improve the manuscript.

**Conflicts of Interest:** The authors declare no conflict of interest.

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
