# Peer review of "Geomorphological Approach to Cliff Instability in Volcanic Slopes: A Case Study from the Gulf of Naples (Southern Italy)"

_geosciences, doi:10.3390/geosciences11070289_

Round 1
Reviewer 1 Report
The paper deals with the susceptibility to collapse of a sector of the Neapolitan rocky coastline made of coherent volcanic tuffs and in-coherent volcanoclastic and pyroclastic deposits, subjected to coastal erosion. The authors analyzed several aspects related to the slope stability in such setting with different approaches, ranging from geomorphological analyses to local geological structural surveying. One of the main outcomes is the proposal of a method for landslide and rock-fall onset susceptibility assessment and mapping in volcanic areas, according to the different behavior of the two main lithologies outcropping.
The authors presented the geological setting in depth, with sufficient references to the available literature. In my opinion, few more references are needed regarding the origin of natural and anthropic caves in the region and regarding the stability of sea-cliffs in similar volcanic setting in presence of caves. Moreover, more details are needed regarding the morphodynamical evolution of coastal caves under the waves’ action.
On the other hand, considering that one of the main outcomes of this study is a landslide susceptibility map, more efforts are needed in the introduction for reporting available studies on slope stability and landslide susceptibility classifications, their possible limits and improvements pursued with this study.
Regarding the methods, their presentation should be revised and improved. The authors should indicate how they measure the indicators with particular reference to the volume of caves, since caves’ volumes depend on the caves’ planimetric and in-depth development, and their effect on cliff stability is not straightforward. The authors should also revise the classes’ ranges for different indicators (e.g. Sea Cliff height I1) since some of them are not represented either in the study area or in the southern Italy. Moreover, the authors should clarify the spatial units to which each class is attributed along with the formulas applied, if any.
Results are presented with sufficient detail and comments, but some improvements could be done to facilitate readers in understanding geological setting and the major outcomes from field structural analyses. Suggestions are reported below.
Following are some point revisions.
Line 61, improper use of “to enhance” instead of “to point out”, to highlight, to underline, etc. Please also check the other occurrences.
Line 72. The sentence could induce misreading since Campi Flegrei are at the border of the Campania Plain which is a wide tectonic depression bordering the Mountain Belt. Please correct.
Line 77. Improper use of “volcanics” instead of volcaniclastic deposits
Line 79. If the “Campi Flegrei” are referred to as a single volcanic area, then the singular form should be used.
Line 85. When reporting the age are the authors referring to the oldest volcanic unit outcropping? The oldest dated one? Please specify.
Line 89. The sentence is quite confusing. The CI is widespread in the Campania Plain even though not outcropping because buried under more recent deposits, e.g. NYT. Please correct.
Line 97. In the are the hydro-magmatic phenomena were frequent but not the only ones, as documented by the ubiquitous fall deposits, frequent lava flows etc. Please correct.
Line 100. When referring to the Monte Nuovo eruption and aging please report references.
Line 105. Report reference for the dynamics of bradyseism.
Line 122. Not clear, probably something is missing? Please check.
Line 124. Also NW-SE striking fractures are common, See Vitale and Isaia (2014).
Line 128. I am not sure that Trentaremi should be translated, and done in the form of Trentaremy.
Line 139. Specify how the structural surveys have been conducted (by means of scan lines? Scan areas? Of which estension?) and how are they representative of such a large area.
Line 163. Are the authors referring just to rock-fall ore more in general to cliff collapse, landslides, rockslides etc? Please check.
Figure 3. Toponyms recalled in the text should be readable.
Line 200. The sentence is not clear. Do the authors mean that in literature, when reported, the unit is grouped with the NYT? Please clarify and report references.
Line 205. The authors should use the term pyroclasts o pyroclastic deposits instead of pyroclastics.
Line 214. Paleosols should be related to the underlying units and not to the overlying ones.
Line 219. Do the authors mean the foothill area? It’s not clear if authors are referring to the stratigraphical or morphological position.
Line 220. Do the authors mean slope debris instead of debris slope?
Figure 4. The stratigraphical relationships are not clear. An attempt to provide a graphical correlation should be made to help readers in understanding the geological complexity and lateral facies variations. The description could be synthesized in the caption/legend area.
Figure 5. The main geographical directions should be reported for each cross section.
Line 245-246. Please replace slope with slope angles.
Line 249. Occurrence and distribution are quite synonyms in this sentence. Moreover, also the type of landslide (Debris avalanches and flows vs rock-falls) is controlled by lithology and slope angles.
Line 255. More comments are needed about the morphodynamical phenomenon related to the enhanced sea erosion occurring in presence of caves. Are natural caves linked to focused sea erosion? How? Do the caves induce and increase in erosion rates? Please clarify.
Figure 8. If the in-dept development of the caves is known, it should be at least drafted in the cross sections.
Line 265. Do the authors mean the lack of vegetation with the expression “relative abundance of vegetation”?
Line 267. Please use “fracture network” instead of “net of fractures”.
Figure 9. Some scalebars or estimation of detached volumes or areas reported in the figures might be helpful.
Line 290. Replace e.g., with i.e.
Line 296. When referring to volcano-tectonics deformation, authors should document their sentence with references or field data.
Line 297-301. Faults and fractures should be distinguished and commented apart.
Figure 11. The figure could be improved with a rose diagram synthesizing the whole field data, along with contour pole plots.
Table 3. Pole to fracture/intersection contour stereo-plots, eventually with analyses of prevailing failure mechanism, could be more helpful than the summarizing table.
Line 334-335. The susceptibility zonation is an interesting outcome therefore it deserves more details about how it has been achieved.
Reviewer 2 Report
This paper investiagtes the problem of cliff stability using a sector of the Posillipo promontory.. The results are very interesting.
Reviewer 3 Report
This is an interesting article about the instability problems of a sea cliff in Southern Italy. The methods are traditional but efficient, and the results are interesting and well presented. However, some minor corrections should be made. I include a copy of the manuscript with comments and suggestions.
Regarding the methodology, some comments should be included about the subjectivity of the procedure followed when estimating instability and weakness factors. The authors must explain how they have selected this combination of variables, discarding others also relevant. A number of similar proposals were already presented by other previous authors for estimating vulnerability factors in cliffs, not cited in the present work. The present proposal must be compared to those previous ones and its advantages be clearly explained.
Regarding the role of fractures in the stability of the cliff, their relationship with the cliff face orientation and slope can be easily achieved graphically by using the stereographic projection to calculate the stability of wedges. I strongly recommend the authors to use this procedure to evaluate cliff stability. You already have the data required for that. Although surely the authors already know how to proceed, a synthesis of the method can be found in Markland (1972) or in Norris and Wyllie (1996).
Finally, English grammar and style must be improved.

Round 2
Reviewer 1 Report
The authors addressed most of the revisions suggested and argued in detail when revision was not possible or declined. Therefore I feel satisfied with the current revised version of the manuscript and I suggest to accept it in the present form.